# Hypnotic enhancement of slow-wave sleep increases sleep-associated hormone secretion and reduces sympathetic predominance in healthy humans

Luciana Besedovsky [1,2✉], Maren Cordi [3], Laura Wißlicen [2], Estefanía Martínez-Albert[2], Jan Born [2] & Björn Rasch [3✉]

Sleep is important for normal brain and body functioning, and for this, slow-wave sleep (SWS), the deepest stage of sleep, is assumed to be especially relevant. Previous studies employing methods to enhance SWS have focused on central nervous components of this sleep stage. However, SWS is also characterized by specific changes in the body periphery, which are essential mediators of the health-benefitting effects of sleep. Here we show that enhancing SWS in healthy humans using hypnotic suggestions profoundly affects the two major systems linking the brain with peripheral body functions, i.e., the endocrine and the autonomic nervous systems (ANS). Specifically, hypnotic suggestions presented at the beginning of a 90-min afternoon nap to promote subsequent SWS strongly increased the release of growth hormone (GH) and, to a lesser extent, of prolactin and aldosterone, and shifted the sympathovagal balance towards reduced sympathetic predominance. Thus, the hypnotic suggestions induced a whole-body pattern characteristic of natural SWS. Given that the affected parameters regulate fundamental physiological functions like metabolism, cardiovascular activity, and immunity, our findings open up a wide range of potential applications of hypnotic SWS enhancement, in addition to advancing our knowledge on the physiology of human SWS.

[1] Institute of Medical Psychology, LMU Munich, Munich, Germany. [2] Institute of Medical Psychology and Behavioral Neurobiology, University of Tübingen, Tübingen, Germany. [3] Department of Psychology, Division of Biopsychology and Methods, University of Fribourg, Fribourg, Switzerland. ✉email: luciana.besedovsky@med.uni-muenchen.de; bjoern.rasch@unifr.ch

Sleep is a fundamental biological process that supports various brain functions, including attention, memory, problem solving, and emotional regulation[1–5]. However, sleep does not only regulate brain activity but also plays a crucial role for several health aspects by modulating processes in the body periphery, such as immune functions, cardiovascular activity, growth, and metabolism[6–9]. The impact of sleep on peripheral functions is principally mediated through two axes, i.e., the endocrine and the autonomic nervous systems (ANS), which can affect any organ and tissue in the body with the respective receptors and therefore have wide-reaching effects[7,8,10]. Slow-wave sleep (SWS), the deepest stage of sleep, is assumed to play a major role in regulating the release of various hormones. Levels of growth hormone (GH), cortisol, prolactin, and aldosterone are temporally strongly associated with electroencephalographic slow-wave activity (SWA), the hallmark of SWS[11–14]. Activity of the ANS is likewise closely associated with SWS, as indicated by a distinct shift in the sympathovagal balance toward reduced sympathetic tone during SWS[15–19].

Hypnosis, which can be defined as a state of consciousness involving focused attention and reduced peripheral awareness[20], has been used in the past as a means to modulate subjective sleep quality and for the treatment of sleep disorders[21–23]. We could previously show that hypnosis can also induce objective changes in sleep architecture. Specifically, listening to hypnotic suggestions to "sleep deeper" while falling asleep increased the time spent in SWS and SWA power, without having any negative side effects[24–26]. However, it is unknown whether hypnotic deepening of sleep also induces changes outside the brain, including modulation of endocrine and ANS activity, although there is evidence that hypnosis can affect body physiology, e.g.,[27,28]. Tackling this question has wide-reaching implications, because these peripheral parameters are essential mediators of the potentially health-benefitting effects of sleep[7,8]. Previous studies employing methods to selectively enhance SWS have largely missed to investigate effects on the peripheral characteristics of this sleep stage, including hormonal secretion and ANS activity[29]. Therefore, in this study, we employed hypnotic suggestions (vs. a neutral control text) for selectively enhancing SWS in hypnotizable healthy young men to investigate whether this simple-to-use intervention invokes conjoint changes in the activity of major endocrine and ANS parameters characterizing natural human SWS. In addition, we measured the number of circulating lymphocytes as an immunological read-out, because these immune cells are reliably affected by sleep and in particular by slow oscillations, which are electroencephalographic slow waves of ≤1 Hz predominating during SWS[29,30].

## Results

Twenty-three healthy young, medium- to high-hypnotizable men participated in two experimental sessions, each encompassing a 90-min afternoon nap in the sleep lab (Fig. 1a). In one session, participants listened to an audio tape that included hypnotic suggestions to sleep deeper (Hypnosis condition) while falling asleep, whereas in the other session, they listened to a neutral control text (Control condition). Sleep was recorded polysomnographically, including electrocardiography for determination of heart-rate variability (HRV). Blood for measurement of hormones was collected 90 min before and at lights off, every 5 min thereafter for 90 min, and again 10 and 20 min after lights on.

**Hypnotic suggestions before sleep increase sleep depth.** The hypnotic suggestions selectively increased the absolute time spent in SWS by 49% ($p = 0.026$) compared to the control condition (Fig. 1b, c and Table 1), which was mainly due to an increased time spent in S4 ($10.45 \pm 2.66$ min vs. $4.61 \pm 1.49$ min; $p = 0.028$). Other sleep stages, total sleep time, wake after sleep onset, sleep onset latency, and subjective sleep quality all remained unchanged (Table 1). Paralleling the increase in SWS duration, the power in the SWA frequency band was increased after listening to hypnotic suggestions compared to the Control condition ($p = 0.046$; Fig. 1b, Supplementary Fig. S1 and Table 1). Similarly, the SWA/beta power ratio, an index of objective sleep quality[31,32], showed a larger shift toward higher amounts of low compared to high frequency power in NonREM sleep after hypnosis compared to the control ($p = 0.016$; Table 1). Thus, hypnotic suggestions before sleep reliably increased sleep depth during the nap.

**Hypnotic deepening of sleep increases hormone secretion.** The hypnotic suggestions strongly increased levels of GH compared to the Control condition during sleep (Condition main effect and Condition × Time interaction: $p < 0.001$; Table 2). The increase started at around 40 min after turning lights off, reaching a maximal difference relative to control of more than 400% after 75 min (Fig. 2a). Furthermore, increases in GH levels from the Hypnosis to the Control condition were correlated with increases in the time spent in SWS ($r = 0.532$, $p = 0.013$; Fig. 2b). Prolactin and aldosterone levels were also increased, albeit to a lesser extent (Condition main effect: $p < 0.001$ and $p = 0.012$, respectively; Fig. 2c, d and Table 2), whereas cortisol levels remained unchanged ($p > 0.259$ for Condition main effect or Condition × Time interaction; Fig. 2e and Table 2).

**Hypnotic suggestions impact measures of the ANS.** The hypnotic suggestions reduced absolute and relative power in the low frequency (LF) range of the HRV spectral analysis during sleep (Condition main effect: $p < 0.001$; Fig. 3a and Supplementary Fig. S2a and Table 2). Whereas relative power in the high frequency (HF) range was significantly increased accordingly (Condition main effect: $p = 0.002$, Fig. 3b), absolute HF power showed a trend toward reduced values in the Hypnosis compared to the Control condition (Condition main effect: $p = 0.096$; Supplementary Fig. S2b and Table 2). Furthermore, increases in SWS duration in the Hypnosis condition with reference to the Control condition were negatively correlated with the relative LF power and positively with the relative HF power ($r = -0.485$, $p = 0.022$ and $r = 0.465$, $p = 0.029$, respectively; Fig. 3e, f). Plasma levels of adrenaline and noradrenaline measured in 30-min intervals during the nap were not significantly changed following the hypnotic suggestions (Condition main effect and Condition × Time interaction: $p > 0.227$; Fig. 3c, d and Table 2). However, increases in SWS duration in the Hypnosis condition were negatively correlated with changes in adrenaline levels on a between-subjects level ($r = -0.468$, $p = 0.033$; Fig. 3g).

**Hypnotic deepening of sleep does not affect lymphocyte counts.** Increasing SWS during the afternoon nap did not significantly affect the number of circulating T cells or B cells in the observed time interval ($p > 0.462$ for Condition main effect or Condition × Time interaction; Table 2 and Supplementary Fig. S3).

**Temporal associations between SWS and peripheral parameters.** To explore temporal associations between the course of SWS and peripheral body changes following the hypnotic suggestions, we performed exploratory cross-correlation analyses for those parameters that were significantly affected by the experimental manipulation. These analyses revealed the highest cross-correlation coefficient between SWS and GH at a time lag of +5

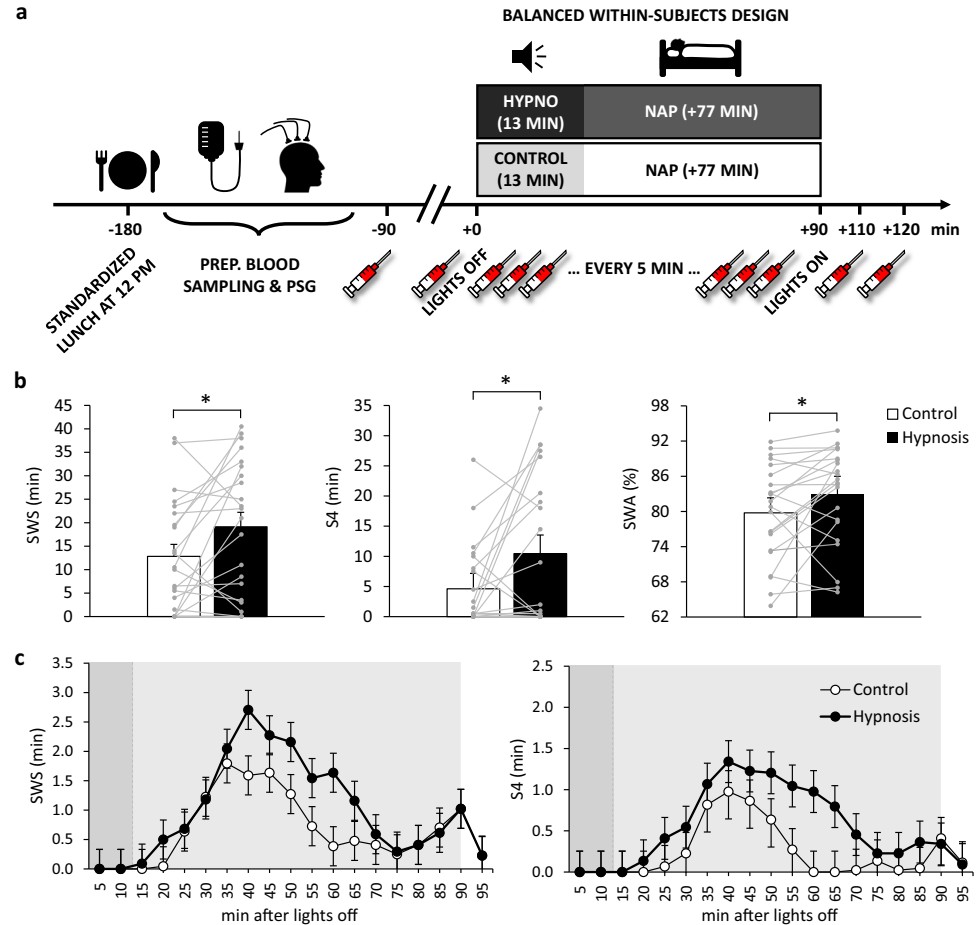

**Fig. 1 Study design and effects of hypnotic suggestions on slow-wave sleep (SWS). a** The experimental sessions started with a standardized lunch at 12 p.m., following which the participants received an intravenous catheter for repeated blood sampling and were prepared for polysomnographic recordings. After lights were turned off for a 90-min nap, the presentation of a 13-min audio tape that included either hypnotic suggestions to sleep deeper (Hypnosis condition) or a neutral control text (Control condition) started. Participants were allowed to fall asleep during presentation of the audio tape and were awakened 77 min after the end of the tape (summing up to a total of 90 min of sleep opportunity). Blood was collected ~90 min and immediately before turning lights off, then every 5 min for 90 min, and 10 and 20 min after turning lights on. Blood was sampled via the intravenous catheter, which was connected to a long tube that enabled blood collection from an adjacent room to avoid disturbing the participants' sleep. PSG polysomnography. **b** Means (±SEM) of time spent in SWS (left) and sleep stage S4 (middle), and relative slow-wave activity (SWA; right) after listening to hypnotic suggestions (black bars) vs. a neutral control text (white bars). **c** Estimated marginal means (±SEM) of minutes spent in SWS and S4 per 5-min bin. Light gray area indicates nap time, darker gray area within nap time indicates duration of the audio tape. *$p < 0.05$ for two-sided paired $t$-tests; see Table 2 for results of linear mixed models analyses; $n = 22$.

to +15 min ($r \geq 0.30$; $p \leq 0.026$) and between SWS and prolactin at a time lag of +15 to +25 min ($r \geq 0.24$; $p \leq 0.006$), indicating that temporal changes in hormone levels lagged behind temporal changes in SWS (Supplementary Fig. S4). No significant cross-correlations were found between SWS and aldosterone. With respect to ANS measures, strongest cross-correlations between temporal changes in SWS and in the relative power of LF and HF were observed at a time lag of +0 min ($r \geq 0.47$; $p < 0.001$; Supplementary Fig. S4). These findings fit with the slower dynamics of hormonal release compared to the fast changes in electro-physiological activities underlying SWS and HRV.

## Discussion

We show here that hypnotic suggestions enhancing SWS during a 90-min nap promote a pattern of endocrine and ANS activity characteristic of natural SWS. Hypnotic suggestions induced a more than fourfold increase in GH levels, smaller increases in prolactin and aldosterone, and a distinct shift of the sympatho-vagal balance toward reduced sympathetic predominance. Hence,

this simple to use hypnotic technique does not only promote central nervous SWS but also impacts the major downstream mediators of the peripheral effects of sleep, demonstrating the physiological significance and validity of this method of SWS enhancement. Since the affected parameters serve various essential physiological functions, including regulation of growth, metabolism, immunity, tissue repair, and cardiovascular activity[7,8,33], the present findings open up a wide range of potential applications of the employed hypnotic suggestions.

Our findings considerably advance our knowledge about the sleep-dependent regulation of hormones and ANS activity. A temporal association between SWS and GH secretion during nocturnal sleep has long been known[12,34–37]. There have been some controversies about the role of SWS in regulating GH secretion, because a dissociation of GH levels and SWS has been observed in certain conditions, such as in night shift workers[38–40]. While GH secretion can occur in the absence of SWS, our study strongly supports a function of SWS-generating mechanisms in GH regulation, which is not dependent on

**Table 1 Comparison of sleep parameters between the Hypnosis and Control condition.**

| | Control | | Hypnosis | | p value | T value | df | Cohen's d |
|---|---|---|---|---|---|---|---|---|
| | **Mean** | **SEM** | **Mean** | **SEM** | | | | |
| **In minutes** | | | | | | | | |
| TST | 71.05 | 4.12 | 74.95 | 4.04 | 0.449 | 0.77 | 21 | 0.20 |
| N1 | 8.82 | 0.89 | 8.07 | 0.88 | 0.535 | 0.63 | 21 | −0.18 |
| N2 | 37.66 | 2.85 | 37.64 | 3.78 | 0.995 | 0.01 | 21 | 0.00 |
| N3 (SWS) | 12.82 | 2.56 | 19.14 | 3.08 | **0.026*** | 2.40 | 21 | 0.48 |
| S3 | 8.20 | 2.03 | 8.68 | 1.65 | 0.721 | 0.36 | 21 | 0.06 |
| S4 | 4.61 | 1.49 | 10.45 | 2.66 | **0.028*** | 2.37 | 21 | 0.58 |
| REMS | 5.07 | 2.06 | 3.61 | 1.22 | 0.484 | 0.71 | 21 | −0.18 |
| WASO | 6.68 | 2.00 | 6.50 | 2.67 | 0.958 | 0.05 | 21 | −0.02 |
| SOL | 10.39 | 2.31 | 9.32 | 1.32 | 0.671 | 0.43 | 21 | −0.12 |
| **In % of TST** | | | | | | | | |
| N1 | 14.77 | 2.56 | 12.25 | 1.70 | 0.359 | 0.94 | 21 | −0.25 |
| N2 | 53.37 | 2.88 | 51.28 | 4.16 | 0.588 | 0.55 | 21 | −0.12 |
| N3 (SWS) | 16.33 | 3.07 | 23.67 | 3.71 | **0.048*** | 2.10 | 21 | 0.46 |
| S3 | 10.27 | 2.30 | 10.94 | 2.08 | 0.701 | 0.39 | 21 | 0.06 |
| S4 | 6.06 | 1.97 | 12.73 | 3.22 | **0.049*** | 2.09 | 21 | 0.53 |
| REMS | 6.10 | 2.42 | 4.39 | 1.44 | 0.488 | 0.71 | 21 | −0.18 |
| WASO | 9.41 | 2.55 | 8.42 | 3.17 | 0.813 | 0.24 | 21 | −0.07 |
| **Spectral power** | | | | | | | | |
| SWA | 79.77 | 1.75 | 82.91 | 1.77 | **0.046*** | 2.13 | 21 | 0.38 |
| SWA/beta | 65.09 | 11.66 | 89.68 | 13.04 | **0.016*** | 2.63 | 21 | 0.42 |
| Subj. sleep quality | 2.25 | 0.17 | 2.11 | 0.12 | 0.292 | 1.08 | 22 | −0.21 |

n = 22 (n = 23 for subjective sleep quality).
df degrees of freedom, TST total sleep time, N1 sleep stage 1, N2 sleep stage 2, N3 encompasses sleep stages S3 and S4 according to the nomenclature of Rechtschaffen and Kales[63], which are also summarized as slow-wave sleep (SWS), REMS Rapid-eye-movement sleep, WASO wake after sleep onset, SOL sleep onset latency, SWA slow-wave activity.
*p < 0.05 for two-sided paired t-tests (shown in bold).

**Table 2 Results of linear mixed models analyses.**

| | Main effect of Condition | | | Main effect of Time | | | Condition x Time interaction | | |
|---|---|---|---|---|---|---|---|---|---|
| | **p value** | **F value** | **df_num, df_den** | **p value** | **F value** | **df_num, df_den** | **p value** | **F value** | **df_num, df_den** |
| **Sleep stages** | | | | | | | | | |
| SWS course | **0.001**** | 40.59 | 1, 777.0 | **<0.001***** | 11.02 | 18, 777.0 | 0.485 | 10.03 | 18, 777.0 |
| S3 course | 0.723 | 0.13 | 1, 777.0 | **<0.001***** | 4.68 | 18, 777.0 | 0.665 | 0.83 | 18, 777.0 |
| S4 course | **<0.001***** | 15.70 | 1, 777.0 | **<0.001***** | 4.91 | 18, 777.0 | 0.718 | 0.79 | 18, 777.0 |
| **Hormone levels** | | | | | | | | | |
| GH | **<0.001***** | 72.10 | 1, 869.6 | **<0.001***** | 7.03 | 21, 866.3 | **<0.001***** | 3.47 | 21, 866.3 |
| Prolactin | **<0.001***** | 63.65 | 1, 866.9 | **<0.001***** | 29.83 | 21, 866.1 | 0.579 | 0.91 | 21, 866.1 |
| Aldosterone | **0.012*** | 6.28 | 1, 825.8 | **<0.001***** | 3.35 | 21, 817.8 | 0.971 | 0.50 | 21, 817.8 |
| Cortisol | 0.260 | 1.27 | 1, 867.6 | **<0.001***** | 20.42 | 21, 865.2 | 0.741 | 0.78 | 21, 865.2 |
| **ANS measures** | | | | | | | | | |
| LF HRV (%) | **<0.001***** | 14.76 | 1, 735.0 | **<0.001***** | 3.14 | 17, 735.0 | 0.860 | 0.64 | 17, 735.0 |
| HF HRV (%) | **0.002**** | 9.66 | 1, 735.0 | **<0.001***** | 3.66 | 17, 735.0 | 0.989 | 0.38 | 17, 735.0 |
| LF HRV (ms²) | **<0.001***** | 14.78 | 1, 735.0 | **0.009**** | 2.01 | 17, 735.0 | 0.759 | 0.74 | 17, 735.0 |
| HF HRV (ms²) | 0.096 | 2.78 | 1, 735.0 | 0.300 | 1.15 | 17, 735.0 | 0.125 | 1.41 | 17, 735.0 |
| Adrenaline | 0.967 | 0.00 | 1, 180.9 | **<0.001***** | 11.14 | 4, 180.3 | 0.877 | 0.30 | 4, 180.4 |
| Noradrenaline | 0.348 | 0.89 | 1, 180.9 | **<0.001***** | 18.23 | 4, 180.3 | 0.228 | 0.30 | 4, 180.4 |
| **Lymphocyte counts** | | | | | | | | | |
| T-cell counts | 0.463 | 0.54 | 1, 90.0 | **0.048*** | 2.74 | 3, 89.2 | 0.926 | 0.16 | 3, 89.2 |
| B-cell counts | 0.684 | 0.18 | 1, 89.3 | **0.001**** | 5.65 | 3, 88.9 | 0.744 | 0.41 | 3, 88.9 |

df_num degrees of freedom of numerator, df_den df of denominator, SWS slow-wave sleep, GH growth hormone, ANS autonomic nervous system, LF HRV low frequency power of heart-rate variability, HF HRV high frequency power of HRV.
*p < 0.05; **p < 0.01; ***p < 0.001 (shown in bold).

nocturnal sleep. The strong correlation between increases in SWS duration and in GH levels in the Hypnosis condition with reference to the Control condition corroborates this notion. Of note, GH levels following SWS enhancement were comparable to peak GH levels normally reached only during nocturnal, SWS-rich sleep[41,42], demonstrating that the hypnotic enhancement of SWS is highly efficient in boosting the secretion of this hormone, independently of the time of day.

The increases in prolactin and aldosterone levels after the hypnosis-induced SWS enhancement are also in line with

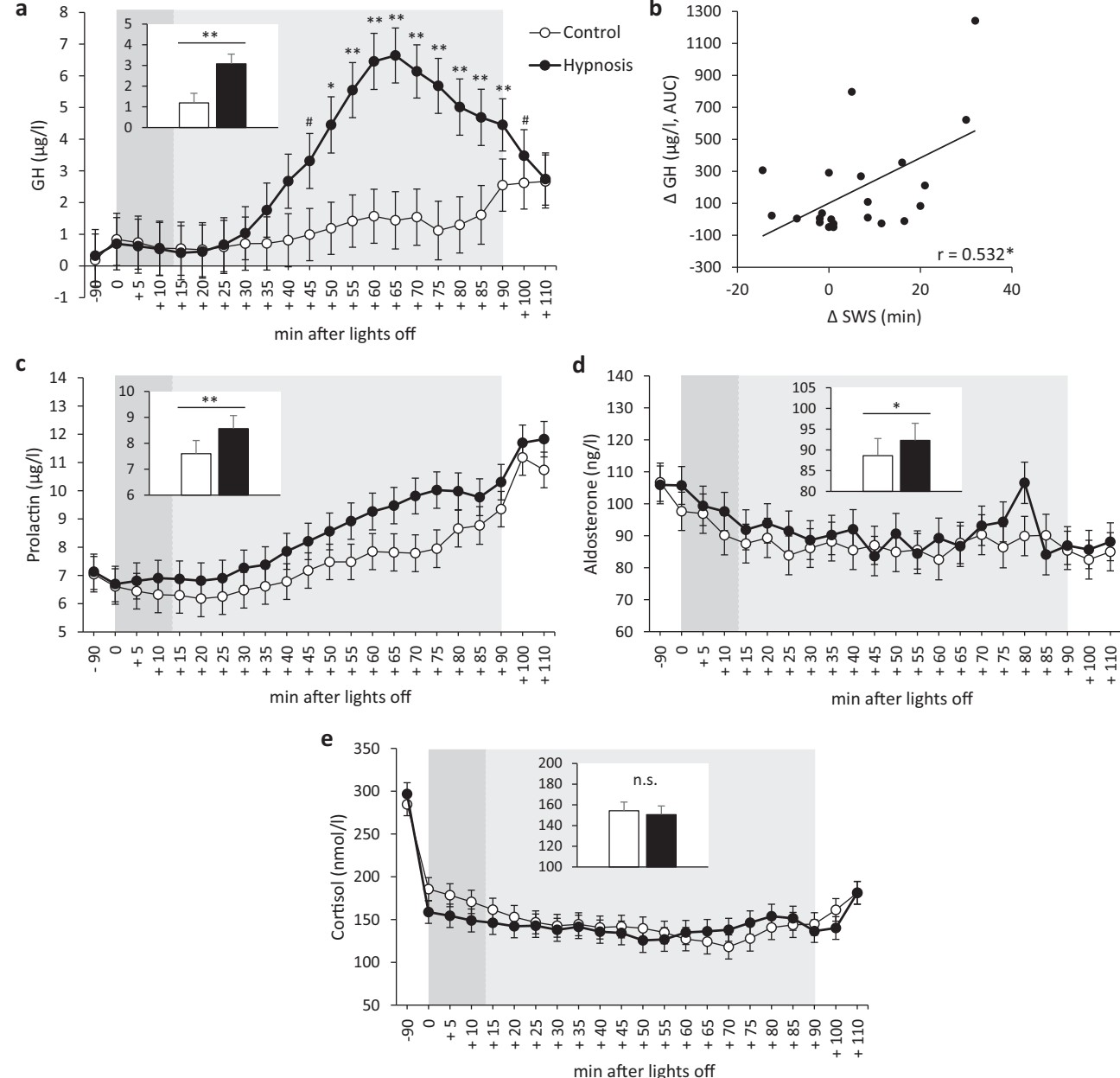

**Fig. 2 Effects of hypnotic suggestions on hormone levels. a** Estimated marginal means (±SEM) of growth hormone levels in the Hypnosis condition (black circles) vs. the Control condition (white circles). **b** Correlations of changes in SWS duration in the Hypnosis relative to the Control condition with changes (in the area under the curve, AUC) of GH levels. **c**–**e** Estimated marginal means (±SEM) of (**c**) prolactin, (**d**) aldosterone, and (**e**) cortisol levels in the Hypnosis condition vs. the Control condition. Light gray area indicates nap time, darker gray area within nap time indicates duration of the audio tape. **p < 0.01, *p < 0.05, #p < 0.10 for the Condition main effect of the linear mixed models analyses shown in the insets, for post hoc pairwise comparisons at single time points if the Condition × Time interaction was significant, and for the correlation coefficient; n.s. not significant; n = 23 for **a**, **c**–**e**; n = 21 for **b** (one participant had to be excluded for the correlational analyses due to too many missing values for the calculation of the AUC and one because of missing polysomnography data).

literature showing a temporal association between SWA and these hormones[11,13]. Our findings support a role of SWS in the regulation of these two hormones, as had been suggested by a previous study using auditory stimulation to enhance electroencephalographic slow oscillations during nocturnal sleep[29]. They further demonstrate that enhancing SWS can be effective in promoting the release of these hormones even during daytime. In contrast to GH, changes in the levels of prolactin and aldosterone between conditions were not correlated with increases in the time spent in SWS. This might be due to the overall

weaker impact of SWS on these two hormones, especially on aldosterone. This hormone is released in a highly pulsatile fashion[13], which might explain why the effect of a SWS manipulation on its secretion is more difficult to detect than for hormones with a steadier secretion, such as GH.

The experimental manipulation did not affect levels of cortisol, despite a previous demonstration of a temporal association between cortisol and SWA[14]. While this might be interpreted as SWS having no effect on cortisol levels, this explanation seems unlikely given previous findings indicating that slow oscillations

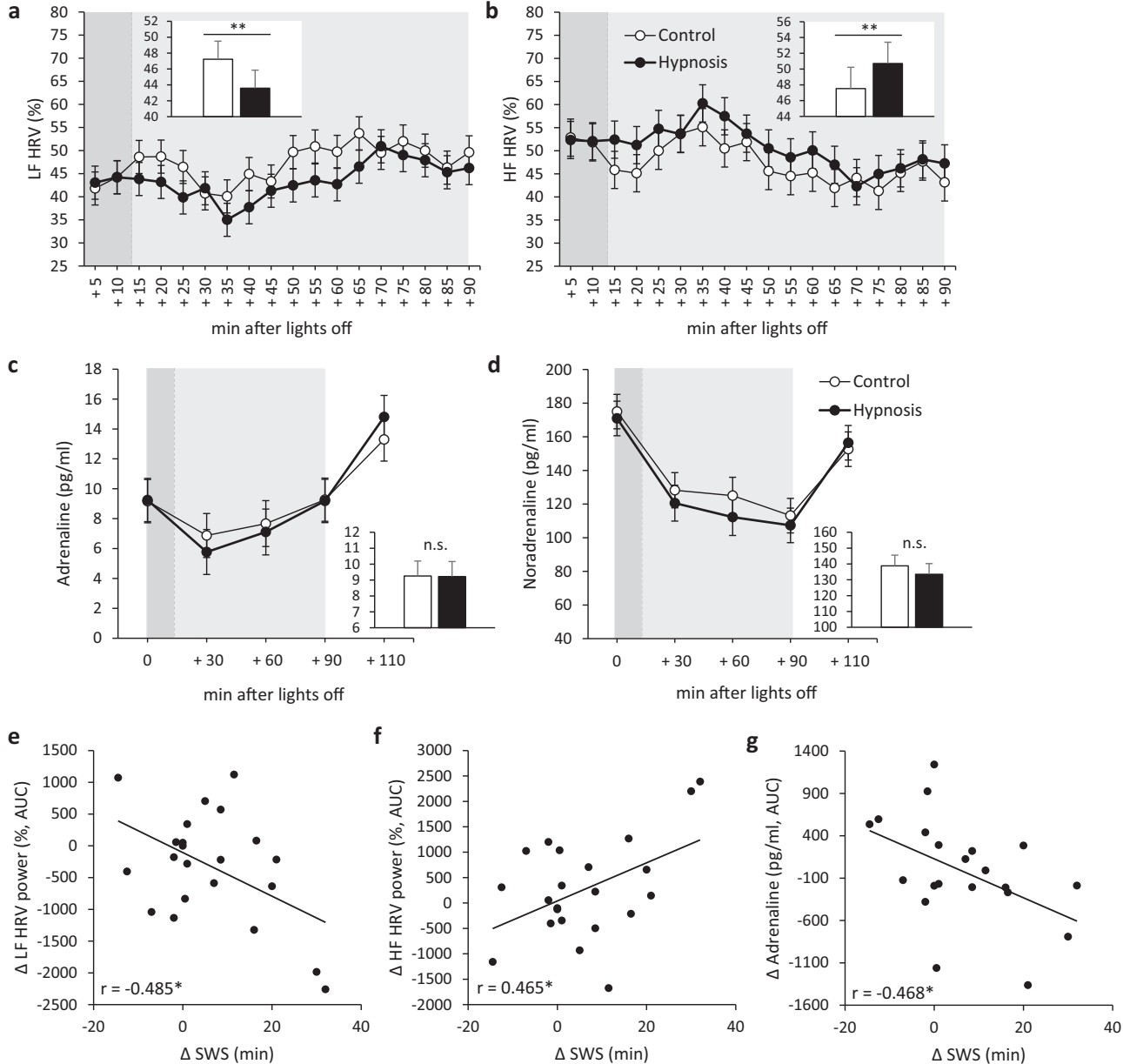

**Fig. 3 Effects of hypnotic suggestions on measures of the autonomic nervous system. a–d** Estimated marginal means (±SEM) of (**a**) relative low frequency power of heart-rate variability (LF HRV), (**b**) relative high frequency power of HRV (HF HRV), (**c**) adrenaline levels, and (**d**) noradrenaline levels in the Hypnosis condition (black circles) vs. the Control condition (white circles). Light gray area indicates nap time, darker gray area within nap time indicates duration of the audio tape. **e–g** Correlations of changes in SWS duration in the Hypnosis relative to the Control condition with (**e**) changes in relative LF HRV power, (**f**) changes in relative HF HRV power, and (**g**) changes (in the area under the curve, AUC) of adrenaline levels. **p < 0.01, *p < 0.05 for the Condition main effect of the linear mixed models analyses shown in the insets and for the correlation coefficients; n.s. not significant; n = 22.

can cause a reduction of cortisol levels and that pituitary-adrenal activity is suppressed during nocturnal SWS[29,43]. The lack of effect in the present study, which focussed on a daytime nap instead of nocturnal sleep, might be due to the tight circadian regulation of this hormone, which is far stronger than for GH and prolactin[44], and might have obscured any less prominent effect of SWS. A similar explanation likely holds true for the lack of effect of SWS enhancement on the number of circulating lymphocytes, which shows a pronounced circadian rhythm that is markedly stronger than the consistently observed impact of nocturnal sleep on these cells[30,45].

Besides influencing endocrine activity, the hypnotic SWS enhancement also affected the ANS, the second major system

linking the brain with the regulation of peripheral functions. The significant reduction in absolute and relative LF power and the corresponding parallel increase in relative HF power indicates a shift of the sympathovagal balance toward reduced sympathetic predominance[46,47]. The findings point toward a contribution of SWS to regulating the sympathovagal balance, which is further supported by the correlation between increases in the time spent in SWS and reductions in relative LF power during the Hypnosis condition with reference to the Control condition. The results are consistent with previous associative findings of reduced LF power during normal SWS compared to other sleep stages[16–18] and with recent findings indicating a strengthening of sleep-autonomic interaction during acoustic stimulation of slow oscillations[48]. The

effects on the ANS further corroborate that the hypnotic SWS enhancement generates a whole-body representation of this sleep stage that is not restricted to the central nervous characteristics of SWS.

We assessed plasma levels of catecholamines as another read-out of ANS activity. Although descriptively adrenaline and nor-adrenaline levels appeared slightly reduced in the Hypnosis condition, this effect was not statistically significant. However, it is to note that for methodological reasons, we were able to measure levels of these hormones only every 30 min, which is probably a too slow sampling rate to detect changes of these hormones having a rather short half-life in plasma of only 1–2 min[49]. Also, effects on local catecholamine release in the tissues are not necessarily captured by such measurements in plasma. Nevertheless, the correlation of increases in SWS dura-tion with decreases in systemic adrenaline levels in the Hypnosis condition (with reference to the Control condition) hints at a possible, although rather weak involvement of SWS in the reg-ulation of this hormone at a systemic level.

Several studies have investigated methods of enhancing SWS, including pharmacological, electrical, magnetic, and acoustic sti-mulation interventions with a focus on central nervous changes[50]. However, SWS is not only characterized by specific brain activity rhythms but comes with typical changes in the body periphery. There is evidence that hypnosis can affect body physiology, e.g.,[27,28], as well as sleep[21–23]; however, previous experiments mainly focussed on subjective sleep quality and none of the previous studies have investigated whether effects of hypnosis on sleep are accompanied by changes in the body periphery. Our present findings reveal hypnotic suggestions enhancing SWS to be a simple to use technique to foster the beneficial effects of SWS on the endocrine system and the ANS. Given that these two systems mediate essential physiological functions of sleep in the entire body, this method of SWS enhancement appears well suited to further investigate the physiological functions of SWS and bears the potential for broad clinical applications, especially in condi-tions of impaired sleep, including e.g., insomnia, depression, and neurodegenerative disorders[51,52]. The effect on GH levels was remarkably large and might prove especially beneficial for situa-tions in which GH levels and SWS amount are low, such as in aged people[53–55] where hypnotic suggestion proved effective in enhancing SWS as well[25]. By allowing a physiological increase of endogenous GH levels our method of SWS enhancement might also add further insights into the current debate about the role of GH in aging[56].

We decided to use a daytime nap instead of nocturnal sleep in the present study to avoid possible ceiling effects, because SWS amount is already very high during nocturnal sleep in our study population of healthy young men. For the same reason, we restricted our sample to male participants, because females gen-erally have even more SWS[57]. We have shown in our previous experiments that the hypnotic suggestions are also effective in enhancing SWS in females during an afternoon nap[24] and also when administered during nocturnal sleep[26]. It would, however, be interesting to investigate whether the hypnotic suggestions are also effective in modulating hormonal and ANS activity during nocturnal sleep as well as in further populations, including females, the elderly, and clinical populations. Investigating effects of SWS enhancement on hormonal release in females would be especially important given known sex differences in the secretion of GH[58]. It also remains to be determined whether stimulating SWS during an afternoon nap has an impact on subsequent nocturnal sleep, e.g., by inducing a homeostatic response leading to reduced SWS propensity and associated hormonal and neu-ronal activity. Furthermore, we cannot entirely exclude that the control text delivered during the control condition disturbed to some extent the quality of subsequent SWS. However, none of the participants had indicated that this had been the case. In our previous study[24], we could further demonstrate that the effect of the hypnotic suggestions was not due to a simple relaxation effect, suggesting that the observed effects were indeed specific to the applied hypnotic suggestions. While hypnotic suggestions to enhance SWS might prove useful for clinical populations with sleep disturbances and metabolic alterations, the effectiveness of this method in enhancing SWS is restricted to medium-to-high-suggestible participants[24], which limits their applicability to a broad population. Interestingly, we have recently shown that enhancing effects of relaxing music on SWS were only evident in low-suggestible participants[59], suggesting that high- vs. low-suggestible people profit from different interventions to enhance SWS. Altogether, while there are still several questions to be investigated in the future, our study demonstrates that hypnotic enhancement of SWS can potently modulate essential peripheral body systems that mediate the health-benefitting effects of sleep.

## Methods
**Participants and experimental procedure**. Twenty-three healthy, German-speaking men (mean age 23.5 years ± 2.71 SD; mean body-mass index: 23.0 kg/m² ± 1.95 SD) participated in this randomized, within-subjects study. Participants had an average habitual sleep duration of 7.6 h (±0.42 SD) and a regular sleep-wake rhythm (both verified by a sleep diary filled in for the 7 days preceding each experimental condition). They did not have any known sleep disturbances, did not take regular naps, were not taking any medication at the time of the experiments, were non-smokers, and refrained from consuming any beverages containing caf-feine in the morning prior to the experimental sessions. Acute and chronic illness was excluded by medical history, physical examination, and routine laboratory investigation. The men were synchronized by daily activities and nocturnal rest. Subjective sleep quality in the night preceding the experimental sessions was on average 1.5 ± 0.45 SD (rated on a scale from 1–6, with higher values indicating worse sleep quality[60]) and did not differ between conditions. The study sample was restricted to male participants because of prominent sex differences in sleep architecture, with males having generally a lower amount of SWS[57], and because the female menstrual cycle can strongly modulate effects of sleep manipulation on hormonal secretion[61]. The study was approved by the Ethics Committee of the University of Tübingen and all participants gave written informed consent.

Before recruitment into the study, volunteers were tested on their suggestibility to hypnosis using the Harvard Group Scale of Hypnotic Susceptibility (HGSHS). Only volunteers with a HGSHS score of 7 or higher were included in the study as previous experiments demonstrated the efficacy of the hypnotic suggestions in increasing SWS only in medium to high-suggestible participants[24]. Eligible volunteers spent an adaptation nap in the laboratory in order to habituate to the experimental setting before the experiment proper. Volunteers then participated in two experimental sessions, a Hypnosis and a Control condition, which were separated by at least 2 weeks. The order of conditions was balanced across participants. Both experimental sessions started at 12 p.m. with a standardized lunch (Fig. 1a). This was followed by the placement of an indwelling catheter for repeated blood sampling and the attachment of electrodes for polysomnographic recordings. When participants were lying in bed (approximately at 3 p.m.), lights were turned off and a tape recording including hypnotic suggestions to "sleep deeper" (Hypnosis condition) or a control text (Control condition) was started. The tape recordings were played over bedside speakers and the duration of the recordings was 13 min. Participants were allowed to fall asleep while listening to the tape and were awakened 90 min after lights had been switched off. Blood for assessment of hormone concentrations was sampled via the intravenous catheter, which was connected to a long thin tube enabling blood collection from an adjacent room without disturbing the participant's sleep. Samples were collected around 90 min and immediately before turning lights off, then every 5 min for 90 min as well as 10 and 20 min after turning lights on. T- and B-lymphocyte counts were assessed in blood samples collected immediately before lights off as well as 45, 90 and 110 min thereafter in a subset of 15 participants.

**Hypnotic and control texts**. The hypnotic tape consisted of a standard hypnotic induction section followed by a hypnotic suggestion section consisting of a metaphor of a fish swimming deeper and deeper into the sea (download available in German at https://www3.unifr.ch/psycho/de/assets/public/Forschungseinheiten/biopsy/hypnose/hypnotic_suggestion_deep_sleep.mp3). The hypnotic text was spoken slowly, with a soft, calm and gentle voice. The control text was about mineral deposits. Its content was taken from a Wikipedia post and adapted in length to the hypnotic text. It was read with a usual every-day voice and normal speed (download (in German) available at https://www3.unifr.ch/psycho/de/assets/public/Forschungseinheiten/biopsy/hypnose/Control_text.mp3).

**Polysomographic recordings and analyses**. The electroencephalogram (EEG) was recorded during sleep from eleven locations (international 10–20 system; F3, Fz, F4, C3, Cz, C4, P3, Pz, P4, O1, and O2) referenced to the average potential from electrodes attached to the mastoids. A ground electrode was attached to the forehead. Additionally, the electrooculogram, electromyogram (EMG) from the chin and electrocardiogram (ECG) were recorded for standard polysomnography. Signals were amplified (Brain Amp, Brain Products, Gilching, Germany) and sampled at a rate of 500 Hz. Sleep stages were determined off-line for subsequent 30-s recording epochs following standard criteria[62]. For a more fine-grained analysis, SWS was further subdivided into stages S3 (delta waves constituting 20–50% of the epoch) and S4 (delta waves constituting >50% of the epoch) according to the criteria of Rechtschaffen and Kales[63].

Power spectra were calculated applying Fast Fourier Transformation. Polysomnographically recorded data were preprocessed using BrainVision Analyzer 2.1 (Brain Products). Data were first filtered using the recommended settings from the American Academy of Sleep Medicine[62] (i.e., low and high cutoff filters: EEG 0.3–35 Hz, EMG 10–100 Hz, ECG 0.3–70 Hz) plus an additional notch filter of 50 Hz. Then all epochs scored as N2 or N3 were selected and a Fast Fourier Transformation was performed on them. First, smaller segments of 4096 data points (~8 s) were created with a 409-point overlap to account for the later applied Hamming window of 10%. These segments were subjected to a semi-automatic artifact rejection, which detected amplitude differences of >600 μV within 200 ms. Afterwards, remaining artificial segments were manually selected and deleted from further analyses. Power (μV²) was analyzed using the full spectrum (0.5–30 Hz) with a resolution of 0.2 Hz and a Hamming window of 10%. Area information of the frequency bands in the SWA range (0.5–4.5 Hz), the beta range (15–30 Hz; for calculation of the SWA/beta power ratio as an index of objective sleep quality), and total power (0.5–50 Hz) during epochs of NonREM sleep was exported. The selection of these parameters was hypothesis-driven. The interested reader is referred to our previous publication for a more fine-grained analysis of polysomnographic data derived from other studies using hypnotic suggestions to deepen sleep[64]. To account for unspecific effects between the two measured sessions, the relative power of each frequency band when setting total power to 100% was calculated. Thus, the reported values represent percentages of total power.

**Hormone and lymphocyte analyses**. GH, prolactin, and cortisol concentrations were assessed in plasma using commercial assays (Siemens Healthcare Diagnostics Inc., Tarrytown, NY, USA), aldosterone levels were measured in plasma by ELISA (DRG Instruments GmbH, Marburg, Germany), and the catecholamines adrenaline and noradrenaline were measured in plasma by HPLC. Sensitivity and intra-assay and interassay variability were as follows: GH: 0.05 μg/l, <6.6%; prolactin: 0.3 μg/l, <4.9%; cortisol: 5.5 nmol/l, <5.5%; aldosterone: 5.7 ng/l, <9.4%), catecholamines: 2 pg/ml, <7.6%. Catecholamine levels were not assessed every 5 min like the other hormones but every 30 min because of the large amount of blood necessary for and the high costs of the HPLC analyses.

Absolute counts of T and B lymphocytes were determined with flow cytometry. In total, 50 μl of an undiluted blood sample were immunostained with anti-CD45, anti-CD3, and anti-CD19 antibodies (at final dilutions of 1:333, 1:333, and 1:50, respectively) in Trucount tubes (BioLegend, San Diego, CA, USA). After 15 min of incubation at room temperature, 0.9 ml of FACS lysing solution (BD Biosciences, San Jose, CA, USA) was added to lyse erythrocytes for 15 min. Samples were then mixed gently, and at least 100,000 CD45+ cells were acquired on a BD LSRFortessa Flow Cytometer (BD Biosciences).

**HRV analyses**. The ECG signal was preprocessed with the Premium version of Kubios HRV, version 3.1.2. A first automatic artifact correction eliminating ectopic beats and artifacts based on dRR series (a time series consisting of the difference between successive RR intervals) was applied to unfiltered data. Data were then segmented into 5 min epochs corresponding to the time points when blood sampling took place. Within those segments, we excluded further visually detected artifacts after, e.g., movements. We used Fast Fourier Transformation to calculate the power densities in the LF (0.04–0.15 Hz) and HF (0.015–0.4 Hz) bands for each 5 min segment. Percent values of LF and HF were calculated using the following formulas: LF [ms²]/total power [ms²] × 100 and HF [ms²]/total power [ms²] × 100, respectively. Note that the normalized LF and HF values are mathematically dependent from each other and therefore do not provide basically different information[46,47].

**Statistics and reproducibility**. Statistical analyses were performed with IBM SPSS Statistics, version 26 (IBM Corp., Armonk, NY, USA). Differences between conditions over time for the course of SWS, S3, and S4, hormone concentrations, HRV parameters, and lymphocyte counts were analyzed using linear mixed models (LMM) including Condition (hypnosis vs. control), Time (the time points of blood collection corresponding to 5-min intervals), and Condition × Time as fixed factors and Participant ID as random factor. Pairwise comparisons within the LMM were performed post hoc to determine which single time points differed significantly between conditions in case the LMM analysis showed a significant Condition × Time interaction. For aldosterone, there was a trend in the baseline value at lights

out (+0 min) in the pairwise comparisons ($p = 0.064$). To exclude that any differences between conditions were a consequence of this slight difference at baseline, we repeated the LMM analysis including the baseline values (i.e., the mean of the two baseline time points −90 and +0 min) as covariate. Because the results only changed marginally, we only show the graph for the baseline-corrected model for aldosterone. Differences in sleep parameters averaged across the entire sleep period were calculated by two-sided paired $t$-tests (some of the sleep parameters were not normally distributed; however, since results of the Wilcoxon signed-rank tests were similar and our sample size was >20, results of the $t$-tests are reported for better comparability of the means with published literature[65]). Correlations of changes in SWS duration from the Control to the Hypnosis condition with changes in hormone levels and HRV measures were calculated using Pearson correlations. For these analyses, the areas under the curve of hormone levels and HRV measures were calculated. A distribution-independent bootstrapping procedure with 10,000 samples was used for the correlational analyses for not normally distributed data (as assessed with the Kolmogorov–Smirnov test). Exploratory cross-correlation analyses were performed for the temporal course of SWS and peripheral body parameters for lags −7 to +7, with each lag corresponding to a 5-min interval. Cross-correlation coefficients were calculated for each individual separately and then z-transformed to calculate significant differences of the coefficients from 0. A $p$ value <0.05 was considered significant. Data are presented as means ± SEM unless otherwise indicated. PSG recordings of one participant and catecholamine levels of another participant could not be analyzed due to insufficient data quality and technical problems, respectively, leading to an $n = 22$ for the respective parameters. The sample size was calculated a priori using power analyses based on previous studies using hypnotic suggestions and pharmacological interventions to increase SWS[24,66]. These studies indicated an effect size of the interventions of $d = 0.59$–0.77. If we consider the smallest effect size, the required sample size for having a power of 0.8 with an alpha of 0.05 and an assumed rho of 0.6 for a two-sided paired $t$ test is 21.

**Reporting summary**. Further information on research design is available in the Nature Research Reporting Summary linked to this article.

## Data availability

Source data for Figs. 1–3 can be found in Supplementary Data 1. All other data are available from the corresponding authors on reasonable request.

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

## Acknowledgements

We thank Miriam Ade for technical help. This work was funded by the European Research Council (ERC) under the European Union's Horizon 2020 research and innovation program 677875 (to B.R.).

## Author contributions

L.B. and B.R. designed the experiments; L.B., L.W., and E.M.A. performed the experiments; L.B. and M.C. analyzed the data; L.B., M.C., J.B., and B.R. interpreted the data; L.B. wrote the paper with contributions from all authors.

## Funding

## Competing interests

The authors declare no competing interests.
