## [Peer Review File · Communications Biology]

Reviewers' comments:

Reviewer #1 (Remarks to the Author):

GENERAL REMARKS

The scientific question presented here is indeed important, as establishing whether hypnotically-extended N3 sleep has the same physiological properties of natural N3 sleep could have dire clinical implications. I do however have some methodological and design observations. I would propose for the authors to be invited to revise and resubmit accordingly.

INTRODUCTION

I find the introduction concise and to the point, as one would expect from a work that follows up on the authors' previous publications. I do find it a bit of a shame that the authors would not present at least briefly the plethora of clinical studies that have been done on hypnosis and sleep as a form of establishing the state of the art. I would also encourage the authors to present some of the previous existing evidence that shows hypnosis can indeed indirectly impact body physiology in a clinical context (such as Anllo et al, 2020). Otherwise, you can also see Terhune et al 2017 for a review.

METHOD & DESIGN

The method is presented in great detail, and the hypnosis recordings are made available. These are excellent practices that help reproducibility, and the authors should be commended on them. I would like to start with a question: did the authors control if sleep onset arrived after the entire hypnosis session was delivered? Do you have any measure of knowing whether some participants fell asleep before hypnosis was delivered, or much after? This is essential, as there could be a source of great individual difference.

I was also very surprised to see that the used sample consisted of only men, which I consider to be a handicapping fault. It should be at least explained why this is the case (if because any methodological constraints or inevitable circumstances). I hope the authors appreciate that this reduces the generalizability of their findings.

Another point I would like to raise is that, while I understand the complexities of having participants come a third night, I would have very much liked to see a third control condition in which a regular nap without any sort of recording is used. While hypnosis induces a form of strong relaxation that could be beneficial for sleep, I don't see the control condition doing the same. Thus, are the effects we are observing right here a result of hypnosis enhancing SWS, or rather the control condition disturbing the quality of SWS? This point should be addressed and justified.

RESULTS

If the entire sleep cycle was recorded polysomnographically, and the analysis was as detailed as the method section shows, it would be nice to see mean differences between control and hypnosis S3/S4 phases expressed on a figure, as a time series or maybe on time-frequency spectra. I also regret to see that there's no topographical plotting of the EEG activity. Finally, while not central to the paper, I would have appreciated some mention to whether hypnosis impacted spindle frequency or K-complex amplitude (again, I understand this study was about N3, but changes on previous stages could be linked to changes later visible in SWS). If the authors think that whatever happened on the other stages was not relevant, then they should justify this stance, or either express lack of in-depth analysis of N1 and N2 sleep as a limitation.

On a separate note:

**Please produce quantitative results and a measure of effect size in every instance where you cite statistically significant differences.

**Please provide justification for sample size, either through simulation-based or regular power analyses. We need to be able to see that your sample has enough power, and therefore your results don't lack validity.

DISCUSSION

The discussion is quite complete, but as with the introduction, I propose that a better effort should be made to connect this research with other examples of hypnosis affecting physiology in general, and sleep in particular. There is a major absence of referencing and connecting the work with actual hypnosis research on the topic

Reviewer #2 (Remarks to the Author):

This is an interesting and well-conducted study, with novel findings related to the peripheral benefits of hypnotic stimulation of deep sleep during an afternoon nap. The manuscript is clearly written. The findings may have clinical implications that will need to be further explored by comparing nighttime sleep versus afternoon napping and examining women (one weakness of the study is that it was conducted exclusively in men), and older adults who are more prone to napping. Another issue worthy of discussion is whether stimulating SWS during an afternoon nap will reduce the propensity for SWS during nighttime sleep and reduce the nocturnal release of GH. Metabolic benefits of afternoon naps have recently been debated and this would be also worthy of discussion.

No information is given about weight and BMI. Since excess adiposity inhibits GH release and overweight/obesity is highly prevalent in modern society (including in Germany), I wonder whether the subjects were selected to be "healthy", i.e. of normal weight. Another missing control is the habitual amount and quality of sleep during the week preceding each study condition, which could have been easily recorded by actigraphy.

An added analysis which might further support a link between the increase in SWS and the increase in GH release would be to show the percentage of each sampling interval spent in S3 and S4 to determine whether the GH increase follows the stimulation of SWS.

There seems to be a discrepancy between text and figure as far as the aldosterone profiles are concerned since the text mentions "showing only the graph for the baseline-corrected model for aldosterone" but the difference in aldosterone levels at time 0 is clearly apparent on the profiles shown in Figure 2D.

When illustrating %LF and %HF in the same figure as amount of SWS, it would be worth on the figure specifying that these spectral estimates relate to HRV, not EEG. Think of readers downloading the slide and getting confused when presenting your work !

Reviewer #3 (Remarks to the Author):

The authors aimed to apply an already proven hypnosis protocol which can increase the time spent in slow-wave stage of sleep (SWS) as well as enhance slow-wave activity, and explore its effects outside the brain. Although slow waves have already been suggested to modulate the endocrine and autonomic nervous systems, this study validates a specific and easy-to-use hypnosis technique which can be used not only to impact the central nervous system and promote SWS, but also to modulate the endocrine and autonomic nervous systems. However, I strongly recommend to add or remark some strengths and limitations of this research:

Comment regarding research strengths and future perspectives:

(1) I encourage authors to dedicate at least a paragraph to discuss future clinical applications. From my point of view, one of the most promising future perspectives in this study is the potential use of the hypnotic protocol in clinical patients. This has been briefly discussed in page 6: "Since the affected parameters serve various essential physiological functions, including regulation of growth, metabolism, immunity, tissue repair, and cardiovascular activity^{7,8,27}, the present findings open up a wide range of potential applications of the employed hypnotic suggestions", and

page 9: "By allowing a physiological increase of endogenous GH levels our method of SWS enhancement might also add further insights into the current debate about the role of GH in aging⁴⁸". I would also add a few examples of how this protocol could potentially be employed in a clinical context, for example, in patients suffering from low sleep quality (by enhancing SWS) or from some metabolic alterations (e.g.: by promoting GH production).

Comments regarding research limitations:

(2) This protocol would only be effective in medium to high-suggestible participants. This is commented at the beginning of the Methods section (page 9): "Only volunteers with a HGSHS score of 7 or higher were included in the study as previous experiments demonstrated the efficacy of the hypnotic suggestions in increasing SWS only in medium to high-suggestible participants". However, it should also be remarked somewhere in the discussion as a limitation of this method, both to increase SWS and to produce peripheral SWS-associated effects.

(3) Similar to comment 3, this protocol is not only restricted to high-suggestible participants, but might also present a different level of effectiveness when implemented in women. All studies supporting the claim that slow-wave sleep modulates the endocrine system have been carried out only in men – at least those cited in this paper (references 11-14, 23, 28-34, 37 in the manuscript), and including this research itself. Given that the secretion of these hormones can present sexual dimorphism, the effects of SWS – and this technique – over the endocrine system may change in women when compared to men. For instance, the first peak in GH during SWS accounts for the majority of the 24-h GH release in men, but not in women – being in this case less pronounced (see Van Cauter & Copinschp, 2000; Jaffe et al., 1998; Veldhuis, 1996). In my view, this should also be remarked in the discussion as a potential limitation of the method.

(4) Future studies could benefit from controlling caffeine ingestion the morning prior to the experiments, since this could affect the quantity of slow-wave sleep and consequently affect the release of some hormones tested in this research (see Landolt, Werth, Borbély & Dijk, 1995; Landolt, Dijk, Gaus & Borbély, 1995; Carrier et al., 2009). It would also be informative to indicate sleep quality and nighttime sleep of participants the night prior to the experiments.

Minor comments:

(5) The exact time at which experimental sessions start – that is to say, in which the standardized lunch starts – is indicated in the Methods section (page 10): "Both experimental sessions started at 12 pm with a standardized lunch". For the sake of convenience, I recommend to include the specific time also in Figure 3.

(6) In addition, the exact time at which the nap starts should be indicated at least in the text for reproducibility reasons – therefore also showing how much time did it take to prepare blood sampling and PSG.

Point-by-point response to reviewers' comments

In detail, the following changes have been introduced regarding the referees' comments (see also tracked changes in the revised manuscript):

Reviewer #1 (Remarks to the Author):

GENERAL REMARKS

The scientific question presented here is indeed important, as establishing whether hypnotically-extended N3 sleep has the same physiological properties of natural N3 sleep could have dire clinical implications. I do however have some methodological and design observations. I would propose for the authors to be invited to revise and resubmit accordingly.

INTRODUCTION

(1) I find the introduction concise and to the point, as one would expect from a work that follows up on the authors' previous publications. I do find it a bit of a shame that the authors would not present at least briefly the plethora of clinical studies that have been done on hypnosis and sleep as a form of establishing the state of the art. I would also encourage the authors to present some of the previous existing evidence that shows hypnosis can indeed indirectly impact body physiology in a clinical context (such as Anllo et al, 2020). Otherwise, you can also see Terhune et al 2017 for a review.

Authors' response:

We thank the reviewer for pointing this out. We have now added literature on effects of hypnosis on sleep and body physiology in the Introduction, lines 44ff:

“Hypnosis, which can be defined as a state of consciousness involving focused attention and reduced peripheral awareness²⁰, has been used in the past as a means to modulate subjective sleep quality and for the treatment of sleep disorders²¹⁻²³. We could previously show that hypnosis can also induce objective changes in sleep architecture. Specifically, listening to hypnotic suggestions to ‘sleep deeper’ while falling asleep increased the time spent in SWS and SWA power, without having any negative side effects²⁴⁻²⁶. However, it is unknown whether hypnotic deepening of sleep also induces changes outside the brain, including modulation of endocrine and ANS activity, although there is evidence that hypnosis can affect body physiology, e.g.,^{27,28}.”

METHOD & DESIGN

(2) The method is presented in great detail, and the hypnosis recordings are made available. These are excellent practices that help reproducibility, and the authors should be commended on them. I would like to start with a question: did the authors control if sleep onset arrived after the entire hypnosis session was delivered? Do you have any measure of knowing whether some participants fell asleep before hypnosis was delivered, or much after? This is essential, as there could be a source of great individual difference.

Authors' response:

The 13-min audio tape presented to the participants was started immediately after lights were turned off. Therefore, the sleep onset latency, which calculates the amount of time needed to fall asleep after lights are turned off, can be used to assess whether the participants fell asleep before

the end of the audio tape. The average sleep onset latency was 10.4 min in the control condition and 9.3 in the hypnosis condition and the difference was not significant. This shows that, on average, participants fell asleep by the end of the 13-min audio tape, independently of the condition. The information on sleep onset latency is given in Table 1 (see below). Of the 22 participants for whom we could analyze the polysomnographic data, 18 fell asleep before the end of the audio tape during the control condition and 17 during the hypnosis condition, showing that there were no large inter-individual differences. We therefore prefer not to discuss this in detail in the manuscript.

Table 1. Comparison of sleep parameters between the Hypnosis and Control condition.

	Control		Hypnosis		P value	T value	df	Cohen's d
	Mean	SEM	Mean	SEM				
In minutes								
TST	71.05	4.12	74.95	4.04	0.449	0.77	21	0.20
N1	8.82	0.89	8.07	0.88	0.535	0.63	21	-0.18
N2	37.66	2.85	37.64	3.78	0.995	0.01	21	0.00
N3 (SWS)	12.82	2.56	19.14	3.08	0.026*	2.40	21	0.48
S3	8.20	2.03	8.68	1.65	0.721	0.36	21	0.06
S4	4.61	1.49	10.45	2.66	0.028*	2.37	21	0.58
REMS	5.07	2.06	3.61	1.22	0.484	0.71	21	-0.18
WASO	6.68	2.00	6.50	2.67	0.958	0.05	21	-0.02
SOL	10.39	2.31	9.32	1.32	0.671	0.43	21	-0.12
In % of TST								
N1	14.77	2.56	12.25	1.70	0.359	0.94	21	-0.25
N2	53.37	2.88	51.28	4.16	0.588	0.55	21	-0.12
N3 (SWS)	16.33	3.07	23.67	3.71	0.048*	2.10	21	0.46
S3	10.27	2.30	10.94	2.08	0.701	0.39	21	0.06
S4	6.06	1.97	12.73	3.22	0.049*	2.09	21	0.53
REMS	6.10	2.42	4.39	1.44	0.488	0.71	21	-0.18
WASO	9.41	2.55	8.42	3.17	0.813	0.24	21	-0.07
Spectral power								
SWA	79.77	1.75	82.91	1.77	0.046*	2.13	21	0.38
SWA/beta	65.09	11.66	89.68	13.04	0.016*	2.63	21	0.42
Subj. sleep quality	2.25	0.17	2.11	0.12	0.292	1.08	22	-0.21

df: degrees of freedom; TST: total sleep time; N1: sleep stage 1; N2: sleep stage 2; N3: encompasses sleep stages S3 and S4 according to the nomenclature of Rechtschaffen and Kales⁶³, which are also summarized as slow-wave sleep (SWS); REMS: Rapid-eye-movement sleep; WASO: wake after sleep onset; SOL: sleep onset latency; SWA: slow-wave activity. **p* < 0.05 for two-sided paired *t* tests; *n* = 22 (*n* = 23 for subjective sleep quality).

(3) I was also very surprised to see that the used sample consisted of only men, which I consider to be a handicapping fault. It should be at least explained why this is the case (if because any methodological constraints or inevitable circumstances). I hope the authors appreciate that this reduces the generalizability of their findings.

Authors' response:

We agree with the reviewer that having limited our sample to male participants reduces the generalizability of our findings. We decided to restrict our sample to male participants for the following reasons: There are prominent sex differences in sleep architecture, with females generally having more SWS (Mong & Cusmano, 2016). We therefore expected that an intervention to increase SWS would have stronger effects in males, allowing to detect also smaller changes in peripheral body functions. Furthermore, it has been shown that the female menstrual cycle can strongly modulate effects of sleep manipulation on hormonal secretion (LeRoux et al., 2014). Given that hormone levels were one of the main outcome variables in the present study, we limited our sample to male participants to reduce inter-individual variability, which could potentially have obscured any smaller effects of the sleep manipulation. In our previous study using hypnotic suggestions during an afternoon nap (Cordi et al., 2014), we demonstrated that the employed hypnotic technique is also effective in increasing SWS in females. However, it remains to be explored whether the effects on peripheral body functions are also evident in females. We have now provided a rationale in the methods section for including only male participants and discuss this as a limitation of our study in the discussion section.

Methods, lines 247ff:

“The study sample was restricted to male participants because of prominent sex differences in sleep architecture, with males having generally a lower amount of SWS⁵⁷, and because the female menstrual cycle can strongly modulate effects of sleep manipulation on hormonal secretion⁶¹.”

Discussion, lines 208ff:

“We decided to use a daytime nap instead of nocturnal sleep in the present study to avoid possible ceiling effects, because SWS amount is already very high during nocturnal sleep in our study population of healthy young men. For the same reason, we restricted our sample to male participants, because females generally have even more SWS⁵⁷. We have shown in our previous experiments that the hypnotic suggestions are also effective in enhancing SWS in females during an afternoon nap²⁴ and also when administered during nocturnal sleep²⁶. It would, however, be interesting to investigate whether the hypnotic suggestions are also effective in modulating hormonal and ANS activity during nocturnal sleep as well as in further populations, including females, the elderly, and clinical populations. Investigating effects of SWS enhancement on hormonal release in females would be especially important given known sex differences in the secretion of GH⁵⁸. It also remains to be determined whether stimulating SWS during an afternoon nap has an impact on subsequent nocturnal sleep...”

(4) Another point I would like to raise is that, while I understand the complexities of having participants come a third night, I would have very much liked to see a third control condition in which a regular nap without any sort of recording is used. While hypnosis induces a form of strong relaxation that could be beneficial for sleep, I don't see the control condition doing the same. Thus, are the effects we are observing right here a result of hypnosis enhancing SWS, or rather the control condition disturbing the quality of SWS? This point should be addressed and justified.

Authors' response:

This is an important point the reviewer is raising. We cannot entirely exclude that the control text delivered during the control condition disturbed to some extent the quality of SWS. However, an unspecific disturbing effect of the control text seems unlikely for several reasons: The duration of the audio tape was only 13 min and it was presented while the participants were falling asleep.

Therefore, for the great majority of the 90-min nap period, the participants were entirely undisturbed. It also seems unlikely that an emotionally neutral text about mineral deposits would affect subsequent SWS. Furthermore, the effect of the hypnosis was specific to SWS: There was no difference in the time spent in the more shallow sleep stages N1 and N2 or in wake after sleep onset. Also, participants did not indicate in the sleep questionnaire that the audio tape subjectively disturbed their subsequent sleep (only two participants indicated that they could not fall asleep immediately because of the audio tape). Therefore, there are no indicators that the control text objectively or subjectively disturbed the participants' sleep. In our previous study, we could further demonstrate that the effect of the hypnotic suggestions was not due to a simple relaxation effect, because a relaxing hypnotic suggestion not containing the suggestion to "sleep deeper" did not affect subsequent SWS. We have now discussed this aspect in our manuscript:

Discussion, lines 220ff:

"Furthermore, we cannot entirely exclude that the control text delivered during the control condition disturbed to some extent the quality of subsequent SWS. However, none of the participants had indicated that this had been the case. In our previous study²⁴, we could further demonstrate that the effect of the hypnotic suggestions was not due to a simple relaxation effect, suggesting that the observed effects were indeed specific to the applied hypnotic suggestions."

RESULTS

(5) If the entire sleep cycle was recorded polysomnographically, and the analysis was as detailed as the method section shows, it would be nice to see mean differences between control and hypnosis S3/S4 phases expressed on a figure, as a time series or maybe on time-frequency spectra. I also regret to see that there's no topographical plotting of the EEG activity. Finally, while not central to the paper, I would have appreciated some mention to whether hypnosis impacted spindle frequency or K-complex amplitude (again, I understand this study was about N3, but changes on previous stages could be linked to changes later visible in SWS). If the authors think that whatever happened on the other stages was not relevant, then they should justify this stance, or either express lack of in-depth analysis of N1 and N2 sleep as a limitation.

Authors' response:

We thank the reviewer for the suggestion to analyze our polysomnographic data in more detail. We have now added a time series of SWS duration in 5-min intervals (corresponding to the sampling rate of the hormonal data) (Fig. 1C). We have now also topographically plotted slow-wave activity (new Fig. S1). As to the suggested analyses on spindle frequency and K complex amplitude, we agree that it is interesting to investigate whether the hypnosis affected these parameters. We are planning a separate publication with a more in-depth analysis of the polysomnographic data (together with results on memory tests, which were also performed in the context of this study). We think that adding such analyses here would draw away the attention from the central question of this manuscript, which is whether hypnotic enhancement of SWS affects peripheral body functions. We have already recently published more fine-grained polysomnographic analyses (including detailed spindle analyses) from our previous studies (see Beck et al., 2021. Nat Sci Sleep). This is now mentioned in the current manuscript:

Figure 1. Study design and effects of hypnotic suggestions on slow-wave sleep (SWS). A: The experimental sessions started with a standardized lunch at 12 pm, following which the participants received an intravenous catheter for repeated blood sampling and were prepared for polysomnographic recordings. After lights were turned off for a 90-min nap, the presentation of a 13-min audio tape that included either hypnotic suggestions to sleep deeper (Hypnosis condition) or a neutral control text (Control condition) started. Participants were allowed to fall asleep during presentation of the audio tape and were awakened 77 min after the end of the tape (summing up to a total of 90 min of sleep opportunity). Blood was collected ~ 90 min and immediately before turning lights off, then every 5 min for 90 min, and 10 and 20 min after turning lights on. Blood was sampled via the intravenous catheter, which was connected to a long tube that enabled blood collection from an adjacent room to avoid disturbing the participants' sleep. PSG: polysomnography. B: Means (\pm SEM) of time spent in SWS (left) and sleep stage S4 (middle), and relative slow-wave activity (SWA; right) after listening to hypnotic suggestions (black bars) versus a neutral control text (white bars). C: Estimated marginal means (\pm SEM) of minutes spent in SWS and S4 per 5-min bin. Light gray area indicates nap time, darker gray area within nap time indicates duration of the audio tape. * $p < 0.05$ for two-sided paired t tests; see Table 2 for results of linear mixed models analyses; $n = 22$.

Supplementary Figure S1. Effect of hypnotic suggestions on slow-wave activity (SWA). Topographical distribution of SWA (0.5–4.5 Hz) during non-rapid eye movement (NREM) sleep after listening to (A) hypnotic suggestions or (B) a neutral control text. SWA is indicated as percent of total power. (C) Percent difference in SWA between the Hypnosis and the Control condition. Black dots represent electrode positions; n = 22.

Methods, lines 303ff: “Area information of the frequency bands in the SWA range (0.5 – 4.5 Hz), the beta range (15 – 30 Hz; for calculation of the SWA/beta power ratio as an index of objective sleep quality), and total power (0.5 – 50 Hz) during epochs of NonREM sleep was exported. The selection of these parameters was hypothesis-driven. The interested reader is referred to our previous publication for a more fine-grained analysis of polysomnographic data derived from other studies using hypnotic suggestions to deepen sleep⁶⁴.”

(6) On a separate note:

***Please produce quantitative results and a measure of effect size in every instance where you cite statistically significant differences.*

Authors’ response:

We appreciate this comment from the reviewer. Effect sizes are shown now for all significant differences when t tests were calculated. For Linear Mixed Models (LMM), we consulted with a statistician who confirmed that there is no established and accepted way of calculating effect sizes for LMM. Given that F values are a function of effect size and sample size, we have now added a table (Table 2) indicating F values and degrees of freedom for all results of LMM. It is possible to calculate a rough estimation of an effect size based on the F values and degrees of freedom using the following formula: $\text{partial } \eta^2 = F \times df_{\text{num}} / (F \times df_{\text{num}} + df_{\text{den}})$. However, given that this is only an approximation that is still difficult to interpret and set in relation to other study designs and statistical models, we would prefer not to add such an estimation of effect size for results of LMM analyses.

Table 2. Results of linear mixed models analyses.

	Main effect of Condition				Main effect of Time				Condition x Time interaction			
	P value	F value	df _{num}	df _{den}	P value	F value	df _{num}	df _{den}	P value	F value	df _{num}	df _{den}
Sleep stages												
SWS course	0.001***	40.59	1, 777.0		<0.001***	11.02	18, 777.0		0.485	10.03	18, 777.0	
S3 course	0.723	0.13	1, 777.0		<0.001***	4.68	18, 777.0		0.665	0.83	18, 777.0	
S4 course	<0.001***	15.70	1, 777.0		<0.001***	4.91	18, 777.0		0.718	0.79	18, 777.0	

Hormone levels									
GH	<0.001***	72.10	1, 869.6	<0.001***	7.03	21, 866.3	<0.001***	3.47	21, 866.3
Prolactin	<0.001***	63.65	1, 866.9	<0.001***	29.83	21, 866.1	0.579	0.91	21, 866.1
Aldosterone	0.012*	6.28	1, 825.8	<0.001***	3.35	21, 817.8	0.971	0.50	21, 817.8
Cortisol	0.260	1.27	1, 867.6	<0.001***	20.42	21, 865.2	0.741	0.78	21, 865.2
ANS measures									
LF HRV (%)	<0.001***	14.76	1, 735.0	<0.001***	3.14	17, 735.0	0.860	0.64	17, 735.0
HF HRV (%)	0.002***	9.66	1, 735.0	<0.001***	3.66	17, 735.0	0.989	0.38	17, 735.0
LF HRV (ms ²)	<0.001***	14.78	1, 735.0	0.009***	2.01	17, 735.0	0.759	0.74	17, 735.0
HF HRV (ms ²)	0.096	2.78	1, 735.0	0.300	1.15	17, 735.0	0.125	1.41	17, 735.0
Adrenaline	0.967	0.00	1, 180.9	<0.001	11.14	4, 180.3	0.877	0.30	4, 180.4
Noradrenaline	0.348	0.89	1, 180.9	<0.001	18.23	4, 180.3	0.228	0.30	4, 180.4
Lymphocyte counts									
T-cell counts	0.463	0.54	1,90.0	0.048*	2.74	3,89.2	0.926	0.16	3,89.2
B-cell counts	0.684	0.18	1,89.3	0.001***	5.65	3,88.9	0.744	0.41	3,88.9

df_{num}: degrees of freedom of numerator; df_{den}: df of denominator; SWS, slow-wave sleep; GH: Growth hormone; ANS: autonomic nervous system; LF HRV: low frequency power of heart-rate variability; HF HRV: high frequency power of HRV.

(7) ***Please provide justification for sample size, either through simulation-based or regular power analyses. We need to be able to see that your sample has enough power, and therefore your results don't lack validity.*

Authors' response:

The sample size was based on previous studies investigating effects of a pharmacological enhancement of SWS on hormonal secretion (Perras et al., 1999) and using hypnotic suggestions to increase SWS during a nap (Cordi et al., 2014; Cordi et al. 2015). For effects of a pharmacological enhancement of SWS on hormonal secretion in healthy young participants, Perras et al. (1999) found an effect size of 0.59 – 0.73 (depending on the hormone investigated). Even if we consider the smaller effect size, the required sample size for having a power of 0.8 is 20. In the study by Cordi et al. (2014) the effect size of hypnotic enhancement of SWS during a nap in healthy young participants was $d = 0.77$. For a power of 0.8, the required sample size is 12. So, our sample size of $n = 23$ had enough power for detecting effects of the hypnotic suggestions on SWS and hormonal secretion. We have now added this aspect to the Materials and Methods section:

Methods, lines 366ff:

“The sample size was calculated using power analyses based on previous studies using hypnotic suggestions and pharmacological interventions to increase SWS^{24,65}.”

DISCUSSION

(8) *The discussion is quite complete, but as with the introduction, I propose that a better effort should be made to connect this research with other examples of hypnosis affecting physiology in general, and sleep in particular. There is a major absence of referencing and connecting the work with actual hypnosis research on the topic*

Authors' response:

We have now added and discussed literature on the effects of hypnosis on sleep and physiology:

Discussion, lines 191ff:

“Several studies have investigated methods of enhancing SWS, including pharmacological, electrical, magnetic, and acoustic stimulation interventions with a focus on central nervous changes⁵⁰.

However, SWS is not only characterized by specific brain activity rhythms but comes with typical changes in the body periphery. There is evidence that hypnosis can affect body physiology, e.g.,^{27,28}, as well as sleep²¹⁻²³; however, previous experiments mainly focussed on subjective sleep quality and none of the previous studies have investigated whether effects of hypnosis on sleep are accompanied by changes in the body periphery.”

Reviewer #2 (Remarks to the Author):

(1) This is an interesting and well-conducted study, with novel findings related to the peripheral benefits of hypnotic stimulation of deep sleep during an afternoon nap. The manuscript is clearly written. The findings may have clinical implications that will need to be further explored by comparing nighttime sleep versus afternoon napping and examining women (one weakness of the study is that it was conducted exclusively in men), and older adults who are more prone to napping. Another issue worthy of discussion is whether stimulating SWS during an afternoon nap will reduce the propensity for SWS during nighttime sleep and reduce the nocturnal release of GH. Metabolic benefits of afternoon naps have recently been debated and this would be also worthy of discussion.

Authors' response:

We thank the reviewer for raising these important points. We decided to restrict our sample to male participants, because females generally have more SWS (Mong & Cusmano, 2016) and because the female menstrual cycle can strongly modulate effects of sleep manipulation on hormonal secretion (LeRoux et al., 2014) (see also our response to question #3 of reviewer 1). We have shown in our previous experiments that the hypnotic suggestions are also effective in enhancing SWS in young and elderly females during an afternoon nap (Cordi et al., 2014) and also when administered during nocturnal sleep (Cordi et al., 2020). However, whether the hypnotic suggestions are also effective in intensifying hormonal and ANS activity during nocturnal sleep as well as in further populations, including females and older adults, still awaits proof.

It is also an important question whether stimulating SWS during a nap will affect subsequent nighttime sleep. Our principal aim was to investigate whether enhancing SWS using hypnotic suggestions would acutely affect endocrine and ANS activity. However, for clinical applications, it is certainly important to investigate whether SWS enhancement during an afternoon nap would induce a compensatory response during subsequent nocturnal sleep, which may cancel out any potentially beneficial effects. We now discuss these aspects in more detail (see below). The literature on whether having regular naps is beneficial for health is indeed inconsistent. We decided to use a daytime nap instead of nocturnal sleep in the present study to avoid possible ceiling effects, because SWS amount is already very high during nocturnal sleep in our study population of healthy young men and because at the time we started the experiments, we did not know whether the hypnosis would also be effective during nocturnal sleep. We think it is beyond the scope to discuss potential beneficial (or disadvantageous) effects of naps, as answering this question was not the purpose of

our study. However, we mention the rationale (and limitation) of using an afternoon nap in the discussion section:

Discussion, lines 208ff:

“We decided to use a daytime nap instead of nocturnal sleep in the present study to avoid possible ceiling effects, because SWS amount is already very high during nocturnal sleep in our study population of healthy young men. For the same reason, we restricted our sample to male participants, because females generally have even more SWS⁵⁷. We have shown in our previous experiments that the hypnotic suggestions are also effective in enhancing SWS in females during an afternoon nap²⁴ and also when administered during nocturnal sleep²⁶. It would, however, be interesting to investigate whether the hypnotic suggestions are also effective in modulating hormonal and ANS activity during nocturnal sleep as well as in further populations, including females, the elderly, and clinical populations. Investigating effects of SWS enhancement on hormonal release in females would be especially important given known sex differences in the secretion of GH⁵⁸. It also remains to be determined whether stimulating SWS during an afternoon nap has an impact on subsequent nocturnal sleep, e.g., by inducing a homeostatic response leading to reduced SWS propensity and associated hormonal and neuronal activity.”

(2) No information is given about weight and BMI. Since excess adiposity inhibits GH release and overweight/obesity is highly prevalent in modern society (including in Germany), I wonder whether the subjects were selected to be "healthy", i.e. of normal weight. Another missing control is the habitual amount and quality of sleep during the week preceding each study condition, which could have been easily recorded by actigraphy.

Authors' response:

The BMI was indeed an inclusion/exclusion criterion. We have now added the information on BMI in the Methods section. Sleep amount and timing during the week preceding each study condition was determined by sleep diaries. In the night previous to the experimental session, a detailed sleep questionnaire was additionally filled in including a measure of subjective sleep quality. We have also added this information.

Methods, lines 237f: “Twenty-three healthy, German-speaking men (mean age 23.5 years \pm 2.71 SD; mean body-mass index: 23.0 kg/m² \pm 1.95 SD) participated in this randomized within-subjects study.”

Methods, lines 238ff and 245ff: “Participants had an average habitual sleep duration of 7.6 hours (\pm 0.42 SD) and a regular sleep-wake rhythm (both verified by a sleep diary filled in for the seven days preceding each experimental condition). ... Subjective sleep quality in the night preceding the experimental sessions was on average 1.5 \pm 0.45 SD (rated on a scale from 1 – 6, with higher values indicating worse sleep quality⁶⁰) and did not differ between conditions.”

(3) An added analysis which might further support a link between the increase in SWS and the increase in GH release would be to show the percentage of each sampling interval spent in S3 and S4 to determine whether the GH increase follows the stimulation of SWS.

Authors' response:

We thank the reviewer for this suggestion. We have now calculated the amount of SWS spent in each sampling interval (shown now in Fig. 1C). We have also performed exploratory cross-correlation

analyses to explore the temporal associations between SWS and peripheral body changes following the hypnotic suggestions (new Fig. S4). While cross-correlation analyses can give hints as to which parameter might drive changes in another parameter, they have to be interpreted with caution and no causal inferences should be made. The lag in the temporal changes of hormone levels might simply reflect the slower dynamics of hormonal release compared to the fast changes in electrophysiological activities underlying SWS and HRV. We mention this point and put the respective figure in the supplements as these analyses can in our opinion only be considered as exploratory.

Results, lines 113ff:

“Temporal associations between SWS and peripheral parameters

To explore temporal associations between the course of SWS and peripheral body changes following the hypnotic suggestions, we performed exploratory cross-correlation analyses for those parameters that were significantly affected by the experimental manipulation. These analyses revealed the highest cross-correlation coefficient between SWS and GH at a time lag of +5 to +15 min ($r \geq 0.30$; $p \leq 0.026$) and between SWS and prolactin at a time lag of +15 to +25 min ($r \geq 0.24$; $p \leq 0.006$), indicating that temporal changes in hormone levels lagged behind temporal changes in SWS (**Fig. S4**). No significant cross-correlations were found between SWS and aldosterone. With respect to ANS measures, strongest cross-correlations between temporal changes in SWS and in the relative power of LF and HF were observed at a time lag of +0 min ($r \geq 0.47$; $p < 0.001$; **Fig. S4**). These findings fit with the slower dynamics of hormonal release compared to the fast changes in electrophysiological activities underlying SWS and HRV.”

Methods, lines 359ff:

“Exploratory cross-correlation analyses were performed for the temporal course of SWS and peripheral body parameters for lags -7 to +7, with each lag corresponding to a 5-min interval. Cross-correlation coefficients were calculated for each individual separately and then z-transformed to calculate significant differences of the coefficients from 0.”

Figure 1. Study design and effects of hypnotic suggestions on slow-wave sleep (SWS). A: The experimental sessions started with a standardized lunch at 12 pm, following which the participants received an intravenous catheter for repeated blood sampling and were prepared for polysomnographic recordings. After lights were turned off for a 90-min nap, the presentation of a 13-min audio tape that included either hypnotic suggestions to sleep deeper (Hypnosis condition) or a neutral control text (Control condition) started. Participants were allowed to fall asleep during presentation of the audio tape and were awakened 77 min after the end of the tape (summing up to a total of 90 min of sleep opportunity). Blood was collected ~ 90 min and immediately before turning lights off, then every 5 min for 90 min, and 10 and 20 min after turning lights on. Blood was sampled via the intravenous catheter, which was connected to a long tube that enabled blood collection from an adjacent room to avoid disturbing the participants' sleep. PSG: polysomnography. B: Means (\pm SEM) of time spent in SWS (left) and sleep stage S4 (middle), and relative slow-wave activity (SWA; right) after listening to hypnotic suggestions (black bars) versus a neutral control text (white bars). C: Estimated marginal means (\pm SEM) of minutes spent in SWS and S4 per 5-min bin. Light gray area indicates nap time, darker gray area within nap time indicates duration of the audio tape. * $p < 0.05$ for two-sided paired t tests; see Table 2 for results of linear mixed models analyses; $n = 22$.

Supplementary Figure S4. Cross correlations for the temporal course of SWS and peripheral body parameters in the Hypnosis condition. Means (±SEM) of cross-correlation coefficients and respective P values for lags -7 to +7. Each lag corresponding to a 5-min interval; n = 22.

(4) There seems to be a discrepancy between text and figure as far as the aldosterone profiles are concerned since the text mentions "showing only the graph for the baseline-corrected model for aldosterone" but the difference in aldosterone levels at time 0 is clearly apparent on the profiles shown in Figure 2D.

Authors' response:

The graph shown for aldosterone is the one with baseline correction. The reason why there is still a slight (but non-significant) difference between conditions at time point +0 min is that we used the mean of both baseline time points (-90 min and +0 min) as a covariate for baseline correction. Using the mean of both baseline time points allows for a more robust correction for baseline differences. We have now clarified in the manuscript how the baseline correction was performed in more detail.

Methods, line 350ff:

"To exclude that any differences between conditions were a consequence of this slight difference at baseline, we repeated the LMM analysis including the baseline values (i.e., the mean of the two baseline time points -90 and +0 min) as covariate."

(5) When illustrating %LF and %HF in the same figure as amount of SWS, it would be worth on the figure specifying that these spectral estimates relate to HRV, not EEG. Think of readers downloading the slide and getting confused when presenting your work !

Authors' response:

We thank the reviewer for making us aware of this potential confusion. We have now clarified in the figures as well as in the figure legends that %LF and %HF refer to HRV.

Figure 3. Effects of hypnotic suggestions on measures of the autonomic nervous system. A-D: Estimated marginal means (\pm SEM) of (A) relative low frequency power of heart-rate variability (LF HRV), (B) relative high frequency power of HRV (HF HRV), (C) adrenaline levels, and (D) noradrenaline levels in the Hypnosis condition (black circles) versus the Control condition (white circles). Light gray area indicates nap time, darker gray area within nap time indicates duration of the audio tape. E-G: Correlations of changes in SWS duration in the Hypnosis relative to the Control condition with (E) changes in relative LF HRV power, (F) changes in relative HF HRV power, and (G) changes (in the area under the curve, AUC) of adrenaline levels. **p < 0.01, *p < 0.05 for the Condition main effect of the

Mixed Models analyses shown in the insets and for the correlation coefficients; n.s., not significant; n = 22.

Supplementary Figure S2. Effects of hypnotic suggestions on measures of absolute low frequency (LF) and high frequency (HF) power of heart-rate variability (HRV). Estimated marginal means (\pm SEM) of absolute LF HRV (left) and absolute HF HRV (right) in the Hypnosis condition (black circles) versus the Control condition (white circles). Light gray area indicates nap time, darker gray area within nap time indicates duration of the audio tape. ** $p < 0.01$, # $p < 0.10$ for the Condition main effect of the Mixed Models analyses shown in the insets; n = 22.

Reviewer #3 (Remarks to the Author):

The authors aimed to apply an already proven hypnosis protocol which can increase the time spent in slow-wave stage of sleep (SWS) as well as enhance slow-wave activity, and explore its effects outside the brain. Although slow waves have already been suggested to modulate the endocrine and autonomic nervous systems, this study validates a specific and easy-to-use hypnosis technique which can be used not only to impact the central nervous system and promote SWS, but also to modulate the endocrine and autonomic nervous systems. However, I strongly recommend to add or remark some strengths and limitations of this research:

Comment regarding research strengths and future perspectives:

(1) I encourage authors to dedicate at least a paragraph to discuss future clinical applications. From my point of view, one of the most promising future perspectives in this study is the potential use of the hypnotic protocol in clinical patients. This has been briefly discussed in page 6: "Since the affected parameters serve various essential physiological functions, including regulation of growth, metabolism, immunity, tissue repair, and cardiovascular activity^{7,8,27}, the present findings open up a wide range of potential applications of the employed hypnotic suggestions", and page 9: "By allowing a physiological increase of endogenous GH levels our method of SWS enhancement might also add further insights into the current debate about the role of GH in aging⁴⁸". I would also add a few examples of how this protocol could potentially be employed in a clinical context, for example, in patients suffering from low sleep quality (by enhancing SWS) or from some metabolic alterations (e.g.: by promoting GH production).

Authors' response:

We thank the reviewer for this suggestion. We mention in the discussion on page 9 that this technique could be beneficial in conditions of impaired sleep and further specify that it could be especially relevant for aged people, who often suffer at the same time not only from low SWS amount but also from low GH levels. We have now added further clinical conditions in which sleep and especially SWS are frequently impaired, including insomnia, depression, and neurodegenerative disorders. We were hesitant to explicitly mention conditions with GH deficiency other than aging, because such conditions are mostly caused by an inability or reduced function of the hypothalamus-pituitary system to produce GH. In such situations, enhancing SWS would likely not lead to an enhanced GH release. We have extended the respective paragraph:

Discussion, lines 199ff:

“Given that these two systems mediate essential physiological functions of sleep in the entire body, this method of SWS enhancement appears well suited to further investigate the physiological functions of SWS and bears the potential for broad clinical applications, especially in conditions of impaired sleep, including e.g. insomnia, depression, and neurodegenerative disorders^{51,52}. The effect on GH levels was remarkably large and might prove especially beneficial for situations in which GH levels and SWS amount are low, such as in aged people⁵³⁻⁵⁵ where hypnotic suggestion proved effective in enhancing SWS as well²⁵. By allowing a physiological increase of endogenous GH levels our method of SWS enhancement might also add further insights into the current debate about the role of GH in aging⁵⁶.”

Comments regarding research limitations:

(2) This protocol would only be effective in medium to high-suggestible participants. This is commented at the beginning of the Methods section (page 9): “Only volunteers with a HGSHS score of 7 or higher were included in the study as previous experiments demonstrated the efficacy of the hypnotic suggestions in increasing SWS only in medium to high-suggestible participants”. However, it should also be remarked somewhere in the discussion as a limitation of this method, both to increase SWS and to produce peripheral SWS-associated effects.

Authors’ response:

This is an important point and we have now added the aspect that the hypnotic suggestion is only effective in medium-to-high-suggestible people to the limitation section.

Discussion, lines 225ff:

“While hypnotic suggestions to enhance SWS might prove useful for clinical populations with sleep disturbances and metabolic alterations, the effectiveness of this method in enhancing SWS is restricted to medium-to-high-suggestible participants²⁴, which limits their applicability to a broad population. Interestingly, we have recently shown that enhancing effects of relaxing music on SWS were only evident in low-suggestible participants⁵⁹, suggesting that high- vs. low-suggestible people profit from different interventions to enhance SWS.”

(3) Similar to comment 3, this protocol is not only restricted to high-suggestible participants, but might also present a different level of effectiveness when implemented in women. All studies supporting the claim that slow-wave sleep modulates the endocrine system have been carried out only in men – at least those cited in this paper (references 11-14, 23, 28-34, 37 in the manuscript), and including this research itself. Given that the secretion of these hormones can present sexual

dimorphism, the effects of SWS – and this technique – over the endocrine system may change in women when compared to men. For instance, the first peak in GH during SWS accounts for the majority of the 24-h GH release in men, but not in women – being in this case less pronounced (see Van Cauter & Copinschp, 2000; Jaffe et al., 1998; Veldhuis, 1996). In my view, this should also be remarked in the discussion as a potential limitation of the method.

Authors' response:

We agree with the reviewer that having limited our sample to male participants reduces the generalizability of our findings. We decided to restrict our sample to male participants for the following reasons: There are prominent sex differences in sleep architecture, with females generally having more SWS (Mong & Cusmano, 2016). We therefore expected that an intervention to increase SWS would have stronger effects in males, allowing to detect also smaller changes in peripheral body functions. Furthermore, it has been shown that the female menstrual cycle can strongly modulate effects of sleep manipulation on hormonal secretion (LeRoux et al., 2014). Given that hormone levels were one of the main outcome variables in the present study, we limited our sample to male participants to reduce inter-individual variability, which could potentially have obscured any smaller effects of the sleep manipulation. Importantly, in our previous study using hypnotic suggestions during an afternoon nap (Cordi et al., 2014), we demonstrated that the employed hypnotic technique is also effective in increasing SWS in females. However, it remains to be explored whether the effects on peripheral body functions are also evident in females. This is even more important given known sex differences in the secretion of GH, as the reviewer points out. We have now provided a rationale in the methods section for including only male participants and discuss this as a limitation of our study in the discussion section.

Methods, lines 247ff:

“The study sample was restricted to male participants because of prominent sex differences in sleep architecture, with males having generally a lower amount of SWS⁵⁷, and because the female menstrual cycle can strongly modulate effects of sleep manipulation on hormonal secretion⁶¹.”

Discussion, lines 208ff:

“We decided to use a daytime nap instead of nocturnal sleep in the present study to avoid possible ceiling effects, because SWS amount is already very high during nocturnal sleep in our study population of healthy young men. For the same reason, we restricted our sample to male participants, because females generally have even more SWS⁵⁷. We have shown in our previous experiments that the hypnotic suggestions are also effective in enhancing SWS in females during an afternoon nap²⁴ and also when administered during nocturnal sleep²⁶. It would, however, be interesting to investigate whether the hypnotic suggestions are also effective in modulating hormonal and ANS activity during nocturnal sleep as well as in further populations, including females, the elderly, and clinical populations. Investigating effects of SWS enhancement on hormonal release in females would be especially important given known sex differences in the secretion of GH⁵⁸. It also remains to be determined whether stimulating SWS during an afternoon nap has an impact on subsequent nocturnal sleep...”

(4) Future studies could benefit from controlling caffeine ingestion the morning prior to the experiments, since this could affect the quantity of slow-wave sleep and consequently affect the release of some hormones tested in this research (see Landolt, Werth, Borbély & Dijk, 1995; Landolt, Dijk, Gaus & Borbély, 1995; Carrier et al., 2009). It would also be informative to indicate sleep quality

and nighttime sleep of participants the night prior to the experiments.

Authors' response:

Participants were asked not to consume any beverages containing caffeine in the morning prior to the experiments. We have added this information in the Methods section. The participants have filled in a questionnaire on their sleep quality in the night prior to the experiments. This information has now also been added.

Methods, lines 238ff and 245ff:

“Participants had an average habitual sleep duration of 7.6 hours (± 0.42 SD) and a regular sleep-wake rhythm (both verified by a sleep diary filled in for the seven days preceding each experimental condition). They did not have any known sleep disturbances, did not take regular naps, were not taking any medication at the time of the experiments, were non-smokers, and refrained from consuming any beverages containing caffeine in the morning prior to the experimental sessions. ... Subjective sleep quality in the night preceding the experimental sessions was on average 1.5 ± 0.45 SD (rated on a scale from 1 – 6, with higher values indicating worse sleep quality⁶⁰) and did not differ between conditions.”

Minor comments:

(5) The exact time at which experimental sessions start – that is to say, in which the standardized lunch starts – is indicated in the Methods section (page 10): “Both experimental sessions started at 12 pm with a standardized lunch”. For the sake of convenience, I recommend to include the specific time also in Figure 3.

Authors' response:

We have now added the specific time of the start of the experimental sessions in the figure showing the experimental design:

Figure 1. Study design and effects of hypnotic suggestions on slow-wave sleep (SWS). A: The experimental sessions started with a standardized lunch at 12 pm, following which the participants received an intravenous catheter for repeated blood sampling and were prepared for polysomnographic recordings. After lights were turned off for a 90-min nap, the presentation of a 13-min audio tape that included either hypnotic suggestions to sleep deeper (Hypnosis condition) or a neutral control text (Control condition) started. Participants were allowed to fall asleep during presentation of the audio tape and were awakened 77 min after the end of the tape (summing up to a total of 90 min of sleep opportunity). Blood was collected ~ 90 min and immediately before turning lights off, then every 5 min for 90 min, and 10 and 20 min after turning lights on. Blood was sampled via the intravenous catheter, which was connected to a long tube that enabled blood collection from an adjacent room to avoid disturbing the participants' sleep. PSG: polysomnography. B: Means (\pm SEM) of time spent in SWS (left) and sleep stage S4 (middle), and relative slow-wave activity (SWA; right) after listening to hypnotic suggestions (black bars) versus a neutral control text (white bars). C: Estimated marginal means (\pm SEM) of minutes spent in SWS and S4 per 5-min bin. Light gray area indicates nap time, darker gray area within nap time indicates duration of the audio tape. * $p < 0.05$ for two-sided paired t tests; see Table 2 for results of linear mixed models analyses; $n = 22$.

(6) In addition, the exact time at which the nap starts should be indicated at least in the text for reproducibility reasons – therefore also showing how much time did it take to prepare blood sampling and PSG.

Authors' response:

We have now added the information that the nap period started approximately at 3 pm.

Methods, lines 261ff:

“When participants were lying in bed (approximately at 3 pm), lights were turned off and a tape recording, including hypnotic suggestions to ‘sleep deeper’ (Hypnosis condition) or a control text (Control condition) was started.”

REVIEWERS' COMMENTS:

Reviewer #1 (Remarks to the Author):

I would like to start by thanking the authors for addressing each of the points I highlighted with care and in detail. I truly believe that this reviewed version is a much improved version of its predecessor, and I sincerely hope the authors agree.

DETAILED RESPONSE:

- (1) My concerns regarding the introduction were addressed. I have no further comment.
- (2) I am satisfied with this answer, but I would appreciate (in an optional capacity, only if you like) if you could mention that everyone fell asleep roughly at the same time.
- (3) I find the explanation sufficient, and the rationale convincing. I would still want a study with a larger gender sample but I understand why this would not be possible here. Causes are clearly presented, and that's enough for me.
- (4) I accept the compromise.
- (5) Excellent additions. Thank you.
- (6) I am aware of the limitations that come with LMM when it comes to effect sizes. I am also aware that there is no "one-fit-all" solution here. I'm afraid this might not be the opportunity to expand on this issue, but given the authors' approach, maybe I would have preferred a forest plot or a table of the model coefficients (and their respective confidence intervals). I leave it up to the editor to determine whether the authors' proposal is appropriate or not. Please do keep in mind that the notion of establishing statistical significance without any index to quantify such effect could be problematic.
- (7) I really like your work and I understand the argument you raise here, but please do one final effort and include the power calculations, specifying that they are a-priori.

I would like to thank the authors for their daring manuscript, and the opportunity of reviewing their work.

Reviewer #3 (Remarks to the Author):

Issues explained in my first review have been addressed and I have no further comments.

Point-by-point response to the reviewer's comments – second round of revision

Comment of Reviewer #1 to previous comment #6:

(6) I am aware of the limitations that come with LMM when it comes to effect sizes. I am also aware that there is no "one-fit-all" solution here. I'm afraid this might not be the opportunity to expand on this issue, but given the authors' approach, maybe I would have preferred a forest plot or a table of the model coefficients (and their respective confidence intervals). I leave it up to the editor to determine whether the authors' proposal is appropriate or not. Please do keep in mind that the notion of establishing statistical significance without any index to quantify such effect could be problematic.

Authors' response:

We agree with the reviewer that it is important to have an index for quantifying the size of an effect. The reason why we are not providing a classical effect size is that there are important limitations to the calculation of effect sizes for linear mixed models. However, we present in Table 2 all F values, which are a function of an effect size and the used sample size. Together with the degrees of freedom, which we also provide in Table 2, in our view, we provide all relevant information that we have to calculate an estimation of an effect size.

Comment of Reviewer #1 to previous comment #7:

(7) I really like your work and I understand the argument you raise here, but please do one final effort and include the power calculations, specifying that they are a-priori.

Authors' response:

We have now included the power calculations and specified that they are a priori.

Methods, lines 370ff:

“The sample size was calculated a priori using power analyses based on previous studies using hypnotic suggestions and pharmacological interventions to increase SWS^{24,65}. These studies indicated an effect size of the interventions of $d = 0.59 - 0.77$. If we consider the smallest effect size, the required sample size for having a power of 0.8 with an alpha of 0.05 and an assumed rho of 0.6 for a two-sided paired t test is 21.”